# Proteome Profiling of the Dura Mater in Patients with Moyamoya Angiopathy

**DOI:** 10.3390/ijms241311194

**Published:** 2023-07-07

**Authors:** Tatiana Carrozzini, Giuliana Pollaci, Gemma Gorla, Antonella Potenza, Nicola Rifino, Francesco Acerbi, Ignazio G. Vetrano, Paolo Ferroli, Anna Bersano, Erica Gianazza, Cristina Banfi, Laura Gatti

**Affiliations:** 1Laboratory of Neurobiology and UCV, Neurology IX Unit, Fondazione IRCCS Istituto Neurologico Carlo Besta, 20133 Milan, Italy; tatiana.carrozzini@istituto-besta.it (T.C.); giuliana.pollaci@istituto-besta.it (G.P.);; 2Department of Pharmacological and Biomolecular Sciences, Università di Milano, 20133 Milan, Italy; 3Department of Neurosurgery, Fondazione IRCCS Istituto Neurologico Carlo Besta, 20133 Milan, Italy; 4Experimental Microsurgical Laboratory, Fondazione IRCCS Istituto Neurologico Carlo Besta, 20133 Milan, Italy; 5Department of Biomedical Sciences for Health, Università di Milano, 20133 Milan, Italy; 6Unit of Functional Proteomics, Metabolomics, and Network Analysis, Centro Cardiologico Monzino IRCCS, 20138 Milan, Italycristina.banfi@cardiologicomonzino.it (C.B.)

**Keywords:** angiopathy, dura mater, filamin A, moyamoya, mass spectrometry, proteome

## Abstract

Moyamoya angiopathy (MMA) is an uncommon cerebrovascular disease characterized by a progressive steno-occlusive lesion of the internal carotid artery and the compensatory development of an unstable network of collateral vessels. These vascular hallmarks are responsible for recurrent ischemic/hemorrhagic strokes. Surgical treatment represents the preferred procedure for MMA patients, and indirect revascularization may induce a spontaneous angiogenesis between the brain surface and dura mater (DM), whose function remains rather unknown. A better understanding of MMA pathogenesis is expected from the molecular characterization of DM. We performed a comprehensive, label-free, quantitative mass spectrometry-based proteomic characterization of DM. The 30 most abundant identified proteins were located in the extracellular region or exosomes and were involved in extracellular matrix organization. Gene ontology analysis revealed that most proteins were involved in binding functions and hydrolase activity. Among the 30 most abundant proteins, Filamin A is particularly relevant because considering its well-known biochemical functions and molecular features, it could be a possible second hit gene with a potential role in MMA pathogenesis. The current explorative study could pave the way for further analyses aimed at better understanding such uncommon and disabling intracranial vasculopathy.

## 1. Introduction

Moyamoya angiopathy (MMA) is an uncommon and disabling intracranial cerebrovascular disease, with variations in prevalence and clinical manifestations across different ethnicities, and which often leads children and young adults to recurrent strokes [1]. MMA is characterized by a progressive steno-occlusive lesion of the terminal part of the internal carotid artery (ICA) and the compensatory development of an unstable network of collateral vessels [2]. These vascular hallmarks are responsible for recurrent ischemic/hemorrhagic strokes, causing severe neurological deficits, physical disabilities and death. Unfortunately, the lack of data on the disease’s pathophysiology and course have limited the development of effective pharmacological treatments aimed at reducing angiopathy progression. Surgical treatment, mainly based on direct and indirect revascularization, is intended to improve cerebral hemodynamics and decrease the pathological collateral network development, and it represents the preferred procedure for patients with MMA [3]. More interestingly, indirect revascularization surgery, based on synangiosis procedures such as encephalodurosynangiosis or encephaloduroarteriomyosynangiosis, is also known to induce spontaneous angiogenesis between the brain surface and the vascularized donor tissues, mainly dura mater (DM) and temporal muscle [4,5]. This angiogenesis is very specific for MMA, but its underlying mechanism is not understood. Very recently, Yamamoto and colleagues investigated the role of the arachnoid membrane in this unique angiogenesis in MMA [6], whereas the function of DM in this mechanism is rather unknown.

During the last years, the role of meninges has been more deeply investigated to unravel its pivotal role in development and brain homeostasis. Meninges emerged as an essential interface within the central nervous system (CNS), being involved into a variety of diseases such as multiple sclerosis, dementia, stroke, viral/bacterial infections, traumatic brain injury and cancer [7]. The DM, the most external meningeal layer, not only provides efficient protection to parenchymal structures but also represents the most important site for cerebrospinal fluid turnover. Additionally, with the discovery of functional lymphatic vessels and numerous immune cells inside the DM, the meningeal sheath emerged as an immunological center, with roles in immune defense, surveillance and homeostasis [7]. Collagen, stromal cells, endothelial cells and abundant immune cells mainly compose the DM. Interestingly, meningeal immunity is also critical to immune responses against stroke and brain injury. Although previous studies have investigated the use of DM-derived cells for personalized medicine applications and disease modeling [8], an exhaustive proteome characterization of human DM is still lacking.

Novel proteomics-based approaches aimed at identifying potential biomarker candidates for diagnostic/prognostic/predictive purposes that possibly reflect specific pathophysiologic aspects of rare cerebrovascular diseases have been performed [9]. Mass spectrometry (MS)-proteome screening of blood and cerebrospinal fluid (CSF) has identified several circulating proteins possibly involved in MMA [10]. However, the lack of information on the protein composition of DM, a tissue very close to the cerebral vessels affected by the disease, has so far prevented us from gaining insights into its role in MMA pathogenesis. Thus, clarifying the cellular/molecular composition of DM may enable a better understanding of this tissue site, with important translational implications for MMA patients’ care.

Overall, the clinical course of MMA is unpredictable and therapeutic decisions are currently based on clinician and surgeon experience. The rarity of MMA and the gaps in knowledge about its pathogenic mechanisms prompted us to apply an explorative but exhaustive molecular approach to characterize the proteome of the DM from MMA patients. Considering DM’s importance in revascularization surgery, we performed a proteomic characterization of DM samples harvested during surgical procedures in a tertiary, national referral center for cerebrovascular diseases, with the final goal of elucidating MMA pathophysiological molecular mechanisms.

## 2. Results

### 2.1. Protein Extraction from Dura Mater (DM) Samples Representing MMA Patient Cohort

This was an explorative study conducted on MMA patients, for whom demographic and clinical features, including cerebrovascular risk factors, were collected.

The MMA diagnosis of the original cohort was performed according to established angiographic diagnostic criteria [1]. Specifically, the presence of stenosis or occlusion at the terminal portion of the ICAs or the proximal segment of the anterior cerebral artery (ACAs) or middle cerebral artery (MCAs) and abnormal vascular networks in the arterial territories near the occlusive or stenotic lesions were considered in the diagnostic criteria. Nevertheless, for all patients, according to the clinical presentation after a basal neuroradiological evaluation with CT or MRI, DSA and, in almost all cases, also advanced analyses were performed to assess the feasibility of and need for surgical revascularization treatment, based on perfusion-optimized CT (PCT), single photon emission CT (SPECT) and positron emission tomography (PET). Patients were considered symptomatic when presenting with TIA, ischemic or haemorrhagic stroke, headache, movement disorders or cognitive disturbances. MMA was classified into the bilateral or unilateral types depending on the number of distal internal carotid arteries observed on DSA (Digital Subtraction cerebral Angiography) [11]. We excluded “moyamoya like” patients according to European Stroke Organisation (ESO) Guidelines [12]; in the whole cohort, possible syndromic diseases were also assessed.

From the original population of 160 white adult patients, consecutively enrolled between November 2014 and December 2022 at the Neurology IX Unit of the Fondazione IRCCS Istituto Neurologico “C. Besta”, a representative subgroup was selected. For the present study we collected samples based on (i) DM availability following neurosurgical procedures and (ii) demographic and clinical features typical of our MMA cohort of patients (80% females, white, mean age of 47 years, bilateral MMA type, clinical presentation of ischemic events) [13,14]. These demographic and clinical characteristics were also representative of the whole MMA population in Western countries [15]. In the patients whose DM was analyzed, chromosome translocations with uncertain clinical significance were found. The median Suzuki stage was 4. Specifically, the clinical and demographical characterizations of the patients whose DM was analyzed are reported in Appendix A.

The DM samples were submitted to an extraction protocol aimed at maximizing the number of extracted proteins. The yields of extracted proteins from 1 mg of tissue were 8.4 ± 0.26 µg.

### 2.2. Mass Spectrometry Proteomic Approach

In order to characterize the protein content of DM, we employed a label-free liquid chromatography-mass spectrometry (LC-MS)-based proteomic approach using the ion mobility separation before peptide fragmentation. For further improving the coverage of the DM proteome, we evaluated the amount of protein identification using a single-dimension LC with a two-dimensional (2D) LC separation. Indeed, we applied 2D separation with a dilution approach to separate the peptide mixture in five fractions at high pH in the first dimension, before using dilution, trapping and separation at a low pH in the second dimension (Figure 1).

In total, we observed an increase from 803 to 1550 different proteins identified in at least one replicate using PLGS 3.3 software, in 1D-LC/MS and 2D-LC/MS, respectively. Considering only proteins identified in two replicates, the number of proteins detected by 2D-LC/MS was 950 and the full list of them is provided in Appendix A.

The addition of a known amount of ADH digest from yeast allowed us to estimate the amount of each identified protein based on the comparison of the intensity of the three most intense peaks of each protein with that of the internal standard. We estimated that the identified proteins were distributed over almost five orders of magnitude by using this approach (Figure 2A). Of note, the 30 most abundant proteins accounted for 50% of the total amount loaded onto column, as reported in Figure 2B.

### 2.3. Gene Ontology Analyses

The gene ontology analysis of the most abundant proteins showed that they were located in the extracellular region (n = 27, *p* = 3.28 × 10^−15^) or exosomes (n = 24, *p* = 6.87 × 10^−17^), and were involved in extracellular matrix organization (n = 15, *p* = 1.98 × 10^−15^) (Figure 3).

Most of the identified proteins were located in the extracellular region, but other brain compartments emerged, such as neuronal cell body (n = 53, *p* = 3.44 × 10^−6^), post-synapse (n = 60, *p* = 9.26 × 10^−6^), pre-synapse (n = 44, *p*= 0.0019), neuron projection cytoplasm (n = 13, *p* = 0.0030) and axon cytoplasm (n = 10, *p* = 0.0061), together with different intracellular organelles (Appendix A).

Gene ontology (GO) analysis in terms of the molecular function of all the identified proteins was performed with the Bingo plugin in Cytoscape (Figure 4) and revealed that most proteins are involved in binding functions, including protein-, nucleotide- and also lipid-binding. Another important represented category was hydrolase activity, including peptidase activity and nucleoside triphosphatase activity. Additionally, proteins endowed with structural activity, enzyme regulator activity, antioxidant activity and transporter activity have been identified, and found to be enriched (Appendix A).

Considering the biological processes in which the identified proteins were involved, we identified more than 1200 enriched processes, ranging from general metabolic processes (cellular component organization, transport, regulation of cellular function and response to stimulus), to processes more specific for this brain compartment, as reported in Table 1, which was extracted from the complete Appendix A.

## 3. Discussion

In the current study, we adopted a label-free quantitative MS-based approach to characterize the proteome of the DM from MMA patients, since a specific characterization of DM was still lacking in MMA. Indeed, the molecular annotation of various brain regions, especially at the protein level, is expected to provide a deeper understanding of the mechanistic basis for diverse functions.

Our preliminary but comprehensive proteomic profile focused—for the first time, as far as we know—on this specific brain tissue because it is closely related to the cerebral vessels affected by MMA, and a spontaneous angiogenesis between the brain surface and the DM was induced using indirect revascularization surgery of MMA steno-occlusive lesions. Thus, clarifying the molecular composition of DM may enable a better understanding of the rare condition, with important translational implications for other cerebrovascular diseases, as well. Moreover, a better understanding of MMA pathogenesis will foster the identification of novel potential therapeutic targets.

The recent technological advancements have allowed cutting-edge proteomic analyses, mainly in neurodegenerative and neuroncological diseases.

On the other hand, many factors have limited (until now) the progress of basic research on MMA, including the lack of appropriate pre-clinical cellular/animal models and the difficulty of obtaining surgical specimens from the ICAs, or related intracranial vessels. Therefore, proteomic investigations reported until now have mainly addressed circulating MMA patients’ proteins/peptides, and not tissue specimens.

A pilot proteomic study identified novel biomarker candidate proteins differentially expressed in MMA CSF by using SELDI-TOF-MS technology [16]. Another proteome analysis highlighted the downregulation of Apolipoprotein-E and Apolipoprotein-J in MMA patients’ CSF [17]. A targeted metabolomics analysis of serum amino acid profiles identified four potential biomarkers for early MMA diagnosis [18]. Serum-derived exosomes, extracted from adult patients diagnosed with pure ischemic or hemorrhagic MMA, revealed disturbed actin dynamics and immunity dysfunction through quantitative proteomics [19]. Very interestingly, a recent transcriptomic profile of ICAs by RNA sequencing identified 133 and 439 sex-specific differentially expressed genes, for male and female MMA patients, respectively, thus highlighting the importance of a “multi-omic” approach in a multi-factorial disease [20].

Another innovative study has examined the pathophysiology of arachnoid membranes in MMA, and underscored the function of fibroblast-derived myofibroblasts that stimulate the production of collagen in the arachnoid membrane and promote specific, spontaneous angiogenesis in MMA [6].

Additional investigations on MMA have focused on meningeal sheets, and specifically on the role of DM in the angiogenic pathways leading to the revascularization of affected cerebral hemispheres, also considering the relative simplicity of obtaining these surgical samples in comparison to intracranial vessels. Indeed, it is really complex to obtain fragments of fragile vessels such as MCA in MMA, and also to avoid further damages of the vascular walls recipient of a bypass, which could determine a failure of the bypass itself, with relevant clinical consequences for the patient. Nevertheless, our understanding of DM largely stems from the use of pre-clinical models, since cells from DM can be isolated, cultured, and utilized for several applications [21,22,23]. A direct, comparative analysis between cells isolated from the dermis and DM, through phenotypic/transcriptomic/genetic profiling, have highlighted the potential limitations of postmortem DM-derived cells for in vitro personalized medicine applications and disease modeling [8].

The angiogenic-driven process derived from DM leads to spontaneous communication between the extracranial and intracranial arteries secondary to wound healing in the granulation tissue [4]. The pressure gradient between the two different arterial systems (respectively, internal and external carotid arteries) allows a constant flow directed to the ICA systems. Experimental studies on revascularization, after indirect anastomosis surgeries in models of chronic cerebral ischemia, have suggested that revascularization required two steps: first, the development of tissues providing vascular beds (angiogenesis), and second, the stimulus of fluid shear stress for artery-to-artery anastomosis (arteriogenesis) [24]. This model was confirmed by the anatomic-pathological analysis of MMA patients: vessels originating from the DM to the brain tissue clearly had the three-layer structure characteristic of arteries, and they were not fragile capillaries [25]. All these findings confirmed the leading role of DM in the revascularization process.

From a methodological point of view, the label-free mass spectrometry-based approach has advantages with respect to labeling approaches, in terms of both sample preparation and instrumental time requirements. Nevertheless, the labeling approach has provided some insights into the potential role of DM proteins in MMA immunogenesis [26].

Our current explorative analysis revealed that the proteome of DM is rather complex, as it contains proteins with several functions spanning hydrolase activity (i.e., peptidase activity and nucleoside triphosphatase activity), antioxidant activity, transporter activity and binding functions, including protein, nucleotide and also lipid binding. Concerning this point, we recently explored the potential connection between lipid metabolism and MMA angiogenic-vasculogenic-pro-inflammatory pathways, because an alteration in selected lipid species could contribute to substantiate the phenotypical disease hallmarks [14]. Moreover, although obesity and dyslipidemia were not considered MMA metabolic anomalies, a potential link between MMA pathogenesis and lipid metabolism has recently been suggested [27].

The 30 most abundant identified proteins, which accounted for 50% of the total protein extract, were located in the extracellular region or exosomes belonging to DM tissue, and were involved in extracellular matrix (ECM) organization. ECM remodeling and angiogenesis regulation were strictly connected to MMA pathophysiology [13]. When the ECM had been degraded and the basement membrane was destroyed, ECs migrated from pre-existing to newly formed MMA collateral vessels. Although it is not completely known which types of cells (e.g., ECs, vascular smooth muscle cells, VSMCs) form the primary lesion in MA, the histopathological features of this rare condition are intimal fibrous thickening and VSMC proliferation in the ICAs [28].

Noticeably, Filamin A (FLNA) was included among the 30 most abundant identified proteins. FLNA is a large actin-binding cytoskeletal protein that is important for cell motility due to stabilizing actin networks and providing scaffolds for multiple cytoskeletal proteins [29].

Interestingly, a C-terminal fragment of FLNA can be cleaved off by calpain to stimulate adaptive angiogenesis under hypoxia, by facilitating the nuclear translocation of HIF-1α and promoting the secretion of VEGF-A [30]. Despite the enhanced activation of calpains detected during ischemic and reperfusion injury, calpain inhibitors in cerebrovascular diseases have not been studied in detail and the mechanisms by which calpain activation promotes cerebral ischemic injury are not defined. Nevertheless, considering the upregulation we found in the DM of our MMA patients (when clinically presenting with ischemic stroke and defective angiogenesis), promising therapeutic options could arise that include this class of small molecule inhibitors.

Undoubtedly, increasing evidence has suggested that FLNA participates in the pathogenesis of cardiovascular, cerebrovascular and respiratory diseases, in which the interaction of FLNA with transcription factors dictate the function of vascular cells [30]. A large body of literature has addressed the morphology and function of filamins in the arterial wall, specifically in VSMCs. This point is of particular interest for MMA pathogenesis, because VSMCs switch from the contractile phenotype to the proliferative, synthetic, migratory and/or macrophage-like phenotypes [31]. Given the impact on VSMC phenotypic switching for MMA, one may speculate that FLNA may be upregulated in DM and can contribute to MMA progression. In a previous proteomic study, FLNA has been found to be downregulated in the media layers isolated from human atherosclerotic arteries, which is not surprising given the completely different pathogenesis that distinguishes the two vascular dysfunctions [32]. FLNA is expressed also by ECs, whereas a lack of FLNA impairs tube formation, vascular remodeling and ECs migration [33].

Due to the wide range of FLNA functions in cell migration/signaling, mutations in the FLNA gene can cause a variable spectrum of developmental malformations. The first identified mutation in the FLNA gene was a null one, resulting in periventricular nodular heterotopia (PVNH) [34]. PVNH is mainly linked to female patients, and it runs a high risk for strokes [34]. Remarkably, FLNA has been recently identified as a possible second-hit gene promoting moyamoya disease-like vascular formation in a PVNH patient, and it was associated with Ring Finger protein 213 (RNF213) p.R4810K variant, the most common MMA polymorphism [35]. It was proposed that the ubiquitin ligase RNF213 targeted FLNA, resulting in the attenuation of the non-canonical WNT/Calcium/NFAT1 pathway in ECs and the disruption of vascular remodeling [36]. This evidence can only strengthen the enormous potentiality deriving from our explorative but exhaustive MS-based approach, performed to characterize the proteome of the DM from MMA patients.

There are several limitations to our study. Firstly, the sample size of MMA patients was relatively small, mostly after patient selection based on DM availability. Thus, our findings should be verified in a larger case setting. Moreover, the study could have suffered from the lack of a proper sex- and age-matched control group of atherosclerotic cerebrovascular disease (ACVD) subjects. Because of ethical issues, it was not possible to obtain dural samples from “healthy” donors. We voluntarily excluded neuro-oncological diseases, because of the possible multiple alterations of neoplastic tissues.

Lastly, we propose to confirm the obtained results about a potential role of FLNA by using suitable cellular models and molecular tools aimed at clarifying the molecular mechanisms of MMA, to fill the gap of knowledge regarding disease pathophysiology.

## 4. Materials and Methods

### 4.1. Human Dura Mater (DM) Samples

This was an explorative study conducted on DM samples from MMA patients diagnosed following the literature criteria and enrolled in the Neurosurgical Department and Neurology IX Unit of the Fondazione IRCCS Istituto Neurologico Carlo Besta (Milan) between November 2014 and October 2022. The entire analyzed DM samples belonging to MMA patients were collected during the neurosurgical procedures, for compliance with the GEN-O-MA project ethical rules. Indeed, according to the indicated project, DM sample collection is allowed only during surgical procedures/interventions performed on MA patients. The full methodology of the enrollment is reported elsewhere [11]. Because of ethical issues, it was not possible to obtain dural samples from “healthy” donors. Upon arrival to the laboratory, a small patch of freshly captured dural tissue, ranging in maximum dimension from 0.5 to 2.5 cm, was dissected and washed in saline solution, then rapidly frozen and banked. The samples were stored at −80 °C until subsequent processing for proteomics.

### 4.2. Ethical Issues

The study design was approved by the Ethics Committee of the Fondazione IRCCS Istituto Neurologico Carlo Besta of Milan (report no. 12, 10 January 2014) and was performed in accordance with the 2013 WMA Declaration of Helsinki. Since it was designed as a purely observational study, patients underwent diagnostic procedures and received therapy according to local practice. Informed written consent for study participation and sample collection from all patients were mandatory for study inclusion. Privacy procedures were applied to protect patients’ personal identities.

### 4.3. Mass Spectrometry Analysis, Data Processing and Statistical Analysis

Since the sample preparation and protein recovery in the biological specimens under analysis still remained the limiting steps in the proteomic workflow, some of the few DM samples were used for the optimization of the protocol for protein extraction. We selected RIPA buffer (170 mg tissue in 1 mL buffer) followed by treatment with a Tissue Lyser instrument (3 min at 25 Hz, Qiagen, Milan, Italy) as the optimal method for an efficient protein recovery with respect to extraction with urea buffer (8 M urea, 2 M thiourea, 4% *w*/*v* CHAPS, 20 mM Tris, and 55 mM dithiotreitol). Indeed, in this last case, protein extract needed to be further treated to eliminate the urea and thiourea, in order to avoid any interference with subsequent protein digestion and liquid chromatography separation. Different DM specimens from two patients were pooled and used for the proteomic analysis, which was performed individually in three technical replicates.

For proteomic analysis, tissues were lysed in RIPA buffer (170 mg in 1 mL buffer) using the Tissue Lyser instrument (3 min at 25 Hz, Qiagen, Milan, Italy). Tissue extracts were precipitated with the protein precipitation kit (Calbiochem, Merk Life Science S.r.l., Milan, Italy). Samples were dissolved in 25 mmol/L NH4HCO3 containing 0.1% RapiGest (Waters Corporation Waters corporation, Milford, MA, USA), sonicated, and centrifuged at 13,000× *g* for 10 min, as previously described [36]. Samples (50 μg of protein) were then incubated for 15 min at 80 °C and reduced with 5 mmol/L DTT at 60 °C for 15 min, followed by carbamidomethylation with 10 mmol/L iodoacetamide for 30 min at room temperature in the darkness. Then, 2.5 μg of sequencing grade trypsin (Promega Italia Srl, Milano, Italy) was added to each sample and it was incubated overnight at 37 °C. After digestion, 2% TFA was added to hydrolyze RapiGest and inactivate trypsin.

Label-free mass spectrometry analysis, LC-MSE, was performed on a hybrid quadrupole-time of flight mass spectrometer (Synapt XS, Waters corporation, Milford, MA, USA) coupled with a UPLC Mclass system with 2D LC Technology and equipped with a nanosource (Waters Corporation, Milford USA) [37]. An orthogonal 2D reversed-phase (RP/RP) approach was used, with the sample (500 ng) loaded at pH 10 and separated into five fractions at pH 10 before analytical separation at pH 2.5 after dilution using a trap column. The first dimension was performed using the XBridge Peptide BEH, C18, 300Å, 5 µm, 1.0 × 50 mm (Waters Corporation, Milford, MA, USA); a 5-step elution gradient was used for the fractionation of the peptide mixture at pH 10 (20 mM ammonium formate in water), and the percentages of ACN in each step were 11.4, 14.7, 17.4, 20.7, and 50.0%, respectively. Peptides eluted from each fraction were on-line diluted and trapped into a Symmetry C18 nanoACQUITY trap column, 100 Å, 5 μm, 180 μm × 2 cm (Waters Corporation, Milford, MA, USA) and subsequently directed to the analytical column HSS T3 C18, 100 Å, 1.7 μm, 75 μm×150 mm (Waters Corporation, Milford, MA, USA) for elution at a flow rate of 300 nL/min by increasing the organic solvent B concentration from 3 to 40% over 90 min, using 0.1% *v*/*v* formic acid in water as reversed phase solvent A, and 0.1% *v*/*v* formic acid in acetonitrile as reversed phase solvent B. Alternatively, samples were separated only in one dimension at pH 3, with the same column and protocol described above for the second dimension. All the assays were made in triplicate and analyzed by LC-MSE, as previously detailed [38], with some modification in ion mobility-enhanced data-independent acquisition (IMS-DIA). The spectral acquisition time in each mode was 0.5 s, with a 0.1 s inter-scan delay. In the low-energy MS mode, the data were collected at a constant collision energy of 6 eV; in the high energy mode, fragmentation was obtained by applying drift-time-specific collision energies. The software PLGS 3.3 was used for protein identification and molar estimation, to a known amount of Alcohol Dehydrogenase from S. Cerevisiae (100 fmol), as previously described [38]. Data were deposited in the ProteomeXchange Consortium via the PRIDE [39] partner repository with the dataset identifier PXD039943.

### 4.4. Gene Ontology Analysis

Proteomics data were analyzed with the Search Tool for the Retrieval of Interacting Genes/Proteins (STRING 10.5) database [40], as previously described [41], to identify enriched gene ontology (GO) terms in the biological process, molecular function or cellular component categories. In particular, we employed the enrichment function of STRING that calculates an enrichment *p*-value based on the Hypergeometric test using the method of Benjamini and Hochberg for the correction of multiple testing (*p* value cut-off of <0.05). Additionally, a GO analysis was performed with all the identified proteins using the Bingo plugin (v 3.0.5) in Cytoscape (v 3.5.1).

## 5. Conclusions

This pilot study provides an optimized workflow for the identification of proteins present in the human DM, which includes sample solubilization, protein digestion, and mass spectrometry identification. Further, it offers a comprehensive proteomic profile of DM in MMA, highlighting proteins not previously identified, such as a cluster of peptides involved in ECM organization. Among the network of analyzed proteins, including hydrolase, antioxidant enzymes, transporter and binding proteins, FLNA emerged as a possible key factor that can contribute to MMA progression, considering that ECM remodeling and angiogenesis regulation are strictly connected to MMA pathophysiology. Finally, the current explorative study could pave the way for further cellular/molecular approaches aimed at better understanding the mechanisms of such an uncommon and disabling intracranial vasculopathy.

## Figures and Tables

**Figure 1 ijms-24-11194-f001:**
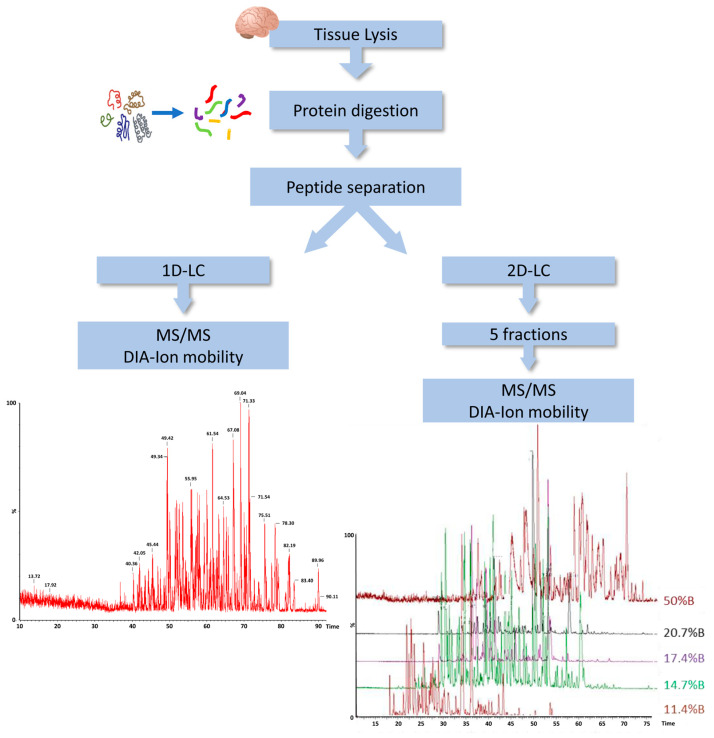
Workflow of DM proteome analysis. DIA, data independent acquisition; LC, liquid chromatography; MS, mass spectrometry; 1D, single-dimensional; 2D, two-dimensional.

**Figure 2 ijms-24-11194-f002:**
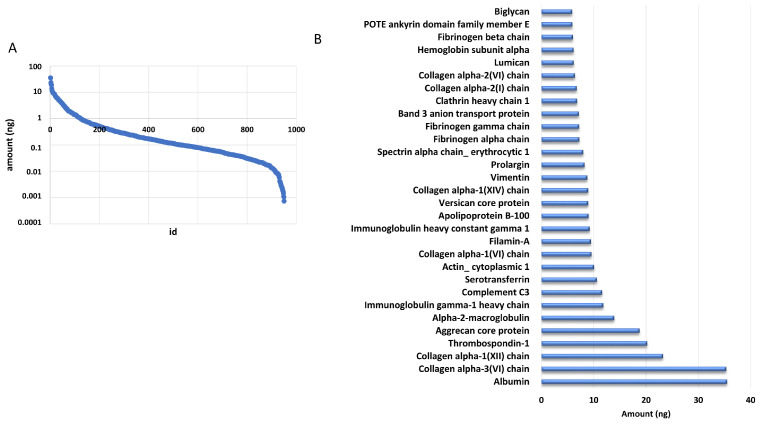
DM proteome distribution. (**A**) Distribution of protein amounts of all the identified proteins. (**B**) Distribution of the most abundant proteins.

**Figure 3 ijms-24-11194-f003:**
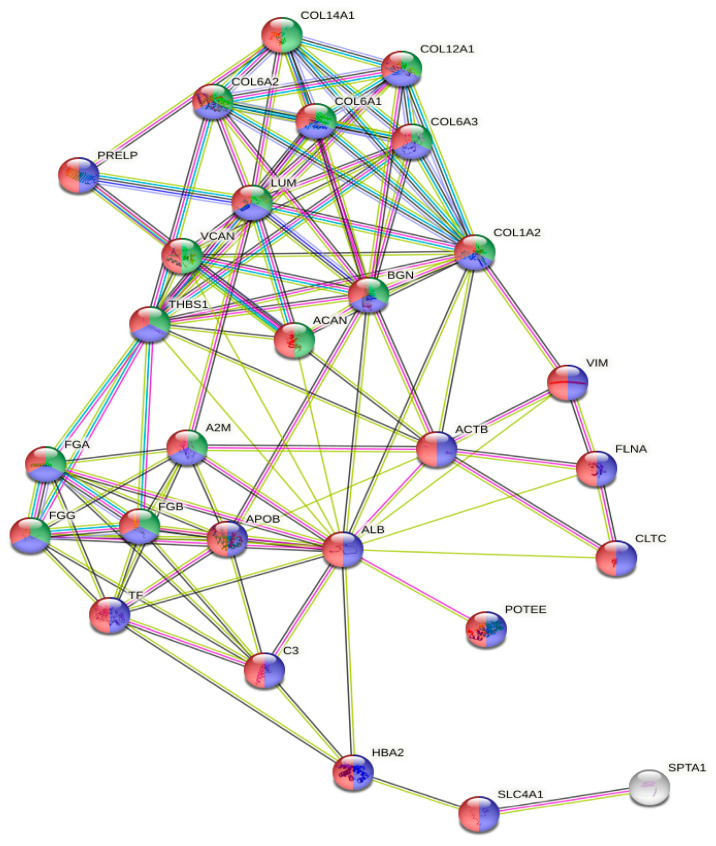
GO analysis of the protein network generated with the 30 most abundant proteins. Red, extracellular region; blue, extracellular exosome; green, extracellular matrix organization.

**Figure 4 ijms-24-11194-f004:**
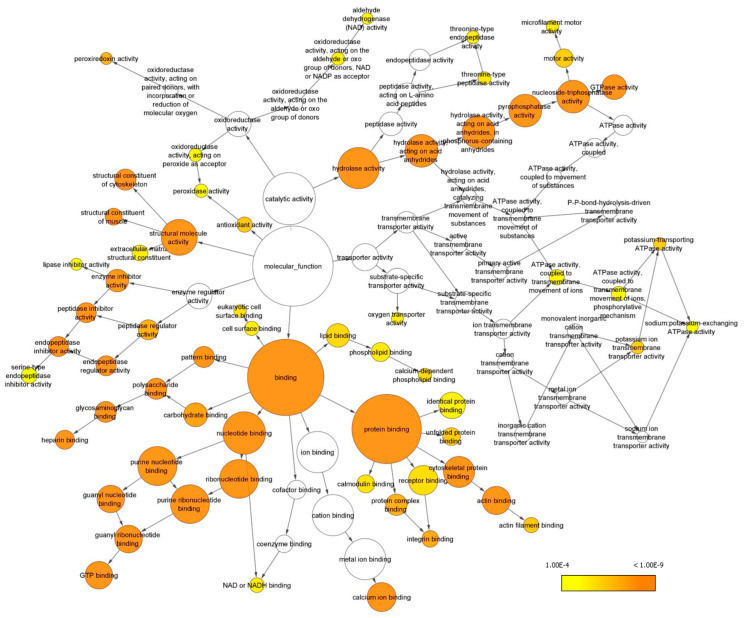
GO analysis of enriched molecular functions. Only GO terms with *p* < 0.0001 according to the Benjamini and Hochberg False Discovery Rate (FDR) correction have been considered and reported. Cytoscape was used to visualize enriched terms resulting from BINGO analysis (see Appendix A).

**Table 1 ijms-24-11194-t001:** List of specific biological processes enriched in the dura mater proteome.

#Term ID	Term Description	Observed Gene Count	Strength	False Discovery Rate
GO:0030901	Midbrain development	17	0.62	7.81 × 10^−5^
GO:0061564	Axon development	40	0.31	0.00074
GO:0014012	Peripheral nervous system axon regeneration	5	1.26	0.0011
GO:0048678	Response to axon injury	11	0.67	0.0013
GO:0031175	Neuron projection development	55	0.24	0.0017
GO:0048812	Neuron projection morphogenesis	43	0.27	0.0022
GO:0008088	Axo-dendritic transport	12	0.59	0.0024
GO:0031102	Neuron projection regeneration	8	0.78	0.0027
GO:0048666	Neuron development	62	0.21	0.0039
GO:0031103	Axon regeneration	7	0.82	0.0040
GO:0022008	Neurogenesis	108	0.15	0.0046
GO:0098930	Axonal transport	10	0.59	0.0071
GO:0008090	Retrograde axonal transport	6	0.83	0.0083
GO:0042063	Gliogenesis	23	0.34	0.0102
GO:0099640	Axo-dendritic protein transport	5	0.92	0.0111
GO:0007409	Axonogenesis	33	0.27	0.0117
GO:0060052	Neurofilament cytoskeleton organization	4	1.09	0.0121
GO:0098974	Postsynaptic actin cytoskeleton organization	5	0.86	0.0175
GO:0099173	Postsynapse organization	12	0.47	0.0178
GO:1901215	Negative regulation of neuron death	21	0.33	0.0180
GO:0033693	Neurofilament bundle assembly	3	1.33	0.0181
GO:0010976	Positive regulation of neuron projection development	26	0.29	0.0191
GO:0099641	Anterograde axonal protein transport	4	0.98	0.0220
GO:0098935	Dendritic transport	4	0.94	0.0280
GO:0098971	Anterograde dendritic transport of neurotransmitter receptor complex	3	1.21	0.0280
GO:0048699	Generation of neurons	96	0.13	0.0317
GO:1990138	Neuron projection extension	9	0.48	0.0449
GO:0061900	Glial cell activation	7	0.57	0.0459
GO:0050808	Synapse organization	24	0.26	0.0479
GO:0007420	Brain development	51	0.17	0.0483

## Data Availability

The data that support the findings of this study are available on request from the corresponding author. The data are not publicly available due to privacy or ethical restrictions. Data collected in the study will be made available using the data repository Zenodo (https://zenodo.org, accessed on 1 April 2023) with restricted access upon request to direzione.scientifica@ccfm.it. Proteomic data are available via ProeomeXchange with identifier PXD039943. Any remaining information can be obtained from the corresponding author upon reasonable request. Anonymized data will be shared by request from a qualified academic investigator for the sole purpose of replicating procedures and results presented in the article and as long as data transfer is in agreement with EU legislation on the General Data Protection Regulation (GDPR, https://gdpr.eu/, accessed on 1 April 2023) and the decision by the Ethical Committee of the Fondazione IRCCS Istituto Neurologico “C. Besta” of Milan, which should be regulated in a material transfer agreement (MTA).

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
