# Peer review of "Proteome Profiling of the Dura Mater in Patients with Moyamoya Angiopathy"

_ijms, 2023, doi:10.3390/ijms241311194_

Round 1
Reviewer 1 Report
Carrozzini et al. investigated the proteomics of DM of MMA patients using LC-MS. They observed proteins in extracellular regions, exosomes, and extracellular matrix organization are the most abundant. Protein binding and hydrolase activity are enriched molecular function. Especially, filamin A might be of interest and discussed in details. I have a few suggestions:
1. The major drawback of this study is the lack of control DM. The identified proteins may be related to DM itself not due to MMA, this is impossible to know without comparing with control DM proteomic data.
2. The other fundamental problem is the use of DM itself, the pathogenesis of MMA begins with microvascular stenosis, followed by angiogenesis in the DM. Hence, if the pathmechanism of MMA is to be studied, endothelium of the vessels is more likely to represent this process. DM is more likely to represent secondary angiogenesis following stenosis not the biology of how MMA occur.
If intracranial vessels samples cannot be obtained, at least the meaning of the finding of DM should be clearly stated in the Discussion.
3. It is quite strange that exact number of patients collected was not described in the results. How many DM samples per patients were analyzed by LC-MS? In total, how many DM samples were studied, were them pooled? How did the experiment replicate?
Minor issues:
1. P.3 Line 96 depending on the number of distal internal?
2. The resolution of the figures was not good.
3. The discussion is a bit too long and should be focusing on discussing the findings of this proteomic study.
Author Response
Comments and Suggestions for Authors
Reviewer 1
We are grateful to the Reviewer for his/her helpful comments and for the careful revision to improve our manuscript. We are confident that the more detailed information we have added will fill the gaps highlighted by the Referee.
- The major drawback of this study is the lack of control DM. The identified proteins may be related to DM itself not due to MMA, this is impossible to know without comparing with control DM proteomic data.
We are grateful to the Referee for the comment that highlighted a true limit of our work, already evidenced by us in Discussion (on page 10, lines 311-315). Actually, the DM samples of MMA patients have been collected during the neurosurgical procedures, for compliance with the GEN-O-MA project ethical rules. The revascularization surgical procedure allows the sampling of a small specimen of DM tissue, considering the role of the dura mater for indirect revascularization (encephaloduroarteriomyosynangiosis), coupled with the direct bypass. Moreover, from an ethical point of view, it is really hard to obtain dural samples from “healthy” donors, considering that we voluntarily excluded neuro-oncological diseases (e.g., meningiomas), because of the possible multiple alterations of neoplastic tissues. Nevertheless, we think that the present manuscript could represent an enrichment to the current literature, to serve as bases for possible future studies, also considering that- at the state of art- a proteomic profile of normal dural tissues is lacking.
We better detailed such issues in Discussion on page 10, lines 315-317 and in Methods on page 10, lines 331-332.
- The other fundamental problem is the use of DM itself, the pathogenesis of MMA begins with microvascular stenosis, followed by angiogenesis in the DM. Hence, if the pathmechanism of MMA is to be studied, endothelium of the vessels is more likely to represent this process. DM is more likely to represent secondary angiogenesis following stenosis not the biology of how MMA occur. If intracranial vessels samples cannot be obtained, at least the meaning of the finding of DM should be clearly stated in the Discussion.
We thank the Reviewer for suggesting this clarification.
MMA is characterized by a progressive steno-occlusive lesion of the terminal part of the internal carotid artery (ICA) and its proximal branches and involves the compensatory development of an unstable network of basal collateral vessels. The cerebral collateral circulatory system is the subsidiary network of vascular channels that stabilize the cerebral blood flow when the principal arteries fail. Since specific pathophysiological mechanisms of MMA have not been elucidated yet, only hypotheses can be raised about the precise timing of such events. The most reliable hypothesis suggested that the new formation of collateral vessels might occur in response to arterial stenosis and/or cerebral ischemia, because of abnormal neo-angiogenesis process. Thus, the role of MMA cerebral microvascular network is still controversial. Nevertheless, we focused our attention on the role of DM because it is involved in the angiogenic pathway leading to revascularization of affected cerebral hemispheres, to provide possible molecular bases for better understanding the role of encephalodurosynangiosis in MA.
Moreover, despite we fully agree with the Reviewer that a proteomic characterization of intracranial vessels could better elucidate some obscure aspects of MMA, it is really complex to obtain fragments of fragile vessels as middle cerebral artery in MMA, also to avoid further damages of the vascular walls recipient of a bypass, that could determine a failure of the bypass itself, with relevant clinical consequences for the patient. We added this part in Discussion on page 8, lines 228-231.
- It is quite strange that exact number of patients collected was not described in the results. How many DM samples per patients were analyzed by LC-MS? In total, how many DM samples were studied, were them pooled? How did the experiment replicate?
We appeciated the Reviewer observation.
Since the sample preparation and protein recovery in the biological specimens under analysis still remain the limiting steps in the proteomic workflow, some of the few DM samples were used for the optimization of the protocol for protein extraction. We selected RIPA buffer (170 mg tissue in 1 ml buffer) followed by treatment with Tissue Lyser instrument (3 minute at 25 Hz), as the optimal method for an efficient protein recovery with respect to extraction with urea buffer (8 M urea, 2 M thiourea, 4% w/v CHAPS, 20 mM Tris, and 55 mM dithiotreitol). Indeed, in this last case, protein extract needs to be further treated to eliminate the urea and thiourea, in order to avoid any interference with subsequent protein digestion and liquid chromatography separation. Different DM specimens from two patients were pooled and used for the proteomic analysis, which was performed individually in three technical replicates. We added this part in Materials and Methods on page 11, lines 346-356.
Minor issues:
- 3 Line 96 depending on the number of distal internal?
We agree with the Reviewer for this precious indication. We corrected the corresponding sentence in the main text (on page 3, line 100-101).
- The resolution of the figures was not good.
We thank the Reviewer for this remark. The Figures embedded in the Word main text for the revision step are characterized by a 600dpi resolution, nonetheless we have provided all the Figures also as separate files (with .TIFF or .jpg format), with an actual good resolution.
- The discussion is a bit too long and should be focusing on discussing the findings of this proteomic study.
We thank the Reviewer for this useful suggestion. We shortened the Discussion section in order to highlight the most relevant findings belonging to our preliminary proteomic study. Specifically, we removed lines 214-224 on page 8, lines 255-258 on page 9, lines 291-294 on page 9, lines 296-301 on page 9, line 319 on page 10, line 327 on page 10, lines 331-341 on page 10. Lines 351-357 on page 10 have been partially moved in materials and methods section on page 11 lines 367-371.

Reviewer 2 Report
The authors present an explorative study of proteomic profiling in MMA. The main limitation is the lack of an appropriate control group, as the authors mention in the limitation. Nevertheless, the paper is still has merit, as a control group would be challenging to obtain. The discussion is well written and the limitations are adequately stated. Some additional information regarding the study cohort would be helpful. I have some other comment below:
Line 27 in abstract should be more objectively stated
Line 86, please deleted the phrase “thanks to the…”
The introduction reads rather long – please try to condense the information a bit to only include what is relevant
Can the authors provide some clarity on how MMA was diagnosed in this cohort, did they need CTA / DSA for diagnosis? Were MM disease and syndrome look at ? how many patients had secondary causes. What was the median Suzuki stage?
Line 99 – why were a subgroup of Caucasian patients analyzed (as opposed to all races)
Adequate
Author Response
Comments and Suggestions for Authors
Reviewer 2
The authors present an explorative study of proteomic profiling in MMA. The discussion is well written and the limitations are adequately stated.
We are grateful to the Reviewer for his/her helpful comments and for the careful revision to improve our manuscript. We are confident that the more detailed information we have added will fill the gaps highlighted by the Referee.
The main limitation is the lack of an appropriate control group, as the authors mention in the limitation. Nevertheless, the paper is still has merit, as a control group would be challenging to obtain.
We are grateful to the Referee for the comment that highlighted a true limit of our work, already evidenced by us in Discussion (on page 10, lines 311-315). Actually, the DM samples of MMA patients have been collected during the neurosurgical procedures, for compliance with the GEN-O-MA project ethical rules. The revascularization surgical procedure allows the sampling of a small specimen of DM tissue, considering the role of the dura mater for indirect revascularization (encephaloduroarteriomyosynangiosis), coupled with the direct bypass. Moreover, from an ethical point of view, it is really hard to obtain dural samples from “healthy” donors, considering that we voluntarily excluded neuro-oncological diseases (e.g., meningiomas), because of the possible multiple alterations of neoplastic tissues. Nevertheless, we think that the present manuscript could represent an enrichment to the current literature, to serve as bases for possible future studies, also considering that- at the state of art- a proteomic profile of normal dural tissues is lacking.
We better detailed such issues in Discussion on page 10, lines 315-317 and in Methods on page 10, lines 331-332.
Some additional information regarding the study cohort would be helpful.
We thank the Reviewer for the suggestion, we agree that some additional information regarding the cohort under study is necessary. We add such information on page 2-3, lines 89-104 and on page 3 lines 112-114, also in response to the other point raised by the Reviewer and below addressed.
I have some other comment below:
Line 27 in abstract should be more objectively stated
As suggested by the Reviewer, previous line 27 in Abstract was rewritten to be more objective: “Among the 30 most abundant proteins, Filamin A is particularly relevant because, considering its well-known biochemical functions and molecular features, it could be a possible second hit gene with a potential role in MMA pathogenesis.”
Line 86, please deleted the phrase “thanks to the…”
Previous Line 86 was deleted, as rightly indicated by Reviewer.
The introduction reads rather long – please try to condense the information a bit to only include what is relevant.
We are grateful to the Referee for his/her useful suggestion. The introduction has been shortened, to better focus the main relevant information. Specifically, we removed previous lines 64-69 on page 2.
Can the authors provide some clarity on how MMA was diagnosed in this cohort, did they need CTA / DSA for diagnosis? Were MM disease and syndrome look at? how many patients had secondary causes. What was the median Suzuki stage?
We thank the Reviewer for suggesting these necessary explanations, since the peculiar features of MMA Caucasian patients as compared to East Asian ones. Indeed, we have recently published the European Stroke Organisation (ESO) Guidelines on Moyamoya angiopathy endorsed by Vascular European Reference Network (VASCERN) [Bersano et al., 2023], which were compiled to assist European clinicians in managing patients with MMA in their decision making. All patients were analyzed by a working group involving neurologists, neurosurgeons, neuroradiologist with high expertice on MA.
MMA diagnosis of the original cohort have been performed according to established angiographic diagnostic criteria as indicated in Ref [1] of the Bibliography. Specifically, the presence of stenosis or occlusion at the terminal portion of the ICAs or the proximal segment of the ACAs or MCAs and abnormal vascular networks in the arterial territories near the occlusive or stenotic lesions were considered in the diagnostic criteria. Nevertheless, all patients, according to the clinical presentation after a basal neuroradiological evaluation with CT or MRI, performed DSA and, in almost all cases, also advanced analyses, to assess the feasibility and need of surgical revascularization treatment, based on perfusion-optimized CT (PCT), single photon emission CT (SPECT), positron emission tomography (PET). Patients were considered symptomatic when presenting with TIA, ischemic or haemorrhagic stroke, headache, movement disorders or cognitive disturbances. We excluded “moyamoya like” patients according to European Stroke Organisation (ESO) Guidelines [Bersano et al., 2023]; in the whole cohort, also possible syndromic diseases have been assessed. In all patients, chromosome translocations with uncertain clinical significance were found. The median Suzuki stage was 4. We added this part on page 2-3, lines 89-104 and on page 3 lines 112-114.
All the individual level information about the patients were summarized in the original Supplementary Table 1.
Line 99 – why were a subgroup of Caucasian patients analysed (as opposed to all races)
We thank the Reviewer for pointed out this confounding sentence; we have corrected the sentence (now on page 3 lines 105-107) and we better explained it in the main text. Indeed, our entire patient cohort is composed of Caucasian subjects and, for the current explorative proteomic study, we selected a small subgroup from this population, essentially based on (i) DM availability and (ii) clinical/demographical features, representative of the whole patient’s cohort.

Round 2
Reviewer 2 Report
The authors have addressed all the concerns appropriately. I have no additional comments.